# Silane-Functionalized Sheep Wool Fibers from Dairy Industry Waste for the Development of Plasticized PLA Composites with Maleinized Linseed Oil for Injection-Molded Parts

**DOI:** 10.3390/polym12112523

**Published:** 2020-10-29

**Authors:** Franciszek Pawlak, Miguel Aldas, Francisco Parres, Juan López-Martínez, Marina Patricia Arrieta

**Affiliations:** 1Faculty of Technology and Chemical Engineering, University of Science and Technology in Bydgoszcz, Seminaryjna 3, PL-85326 Bydgoszcz, Poland; 2Instituto de Tecnología de Materiales, Universitat Politècnica de València, Plaza Ferrándiz y Carbonelle, 03801 Alcoy-Alicante, Spain; fraparga@dimm.upv.es (F.P.); jlopezm@mcm.upv.es (J.L.-M.); 3Departamento de Ciencia de Alimentos y Biotecnología, Facultad de Ingeniería Química y Agroindustria, Escuela Politécnica Nacional, Ladrón de Guevera E11-253, Quito 170517, Ecuador; 4Departamento de Ingeniería Química y del Medio Ambiente, Escuela Politécnica Superior de Ingenieros Industriales, Universidad Politécnica de Madrid (ETSII-UPM), Calle José Gutiérrez Abascal 2, 28006 Madrid, Spain; 5Grupo de Investigación—Polímeros, Caracterización y Aplicaciones (POLCA), 28006 Madrid, Spain

**Keywords:** PLA, maleinized linseed oil, sheep wool fiber, injection molding, silane coupling agents

## Abstract

Poly(lactic acid) (PLA) was plasticized with maleinized linseed oil (MLO) and further reinforced with sheep wool fibers recovered from the dairy industry. The wool fibers were firstly functionalized with 1 and 2.5 phr of tris(2-methoxyethoxy)(vinyl) (TVS) silane coupling agent and were further used in 1, 5, and 10 phr to reinforce the PLA/MLO matrix. Then, the composite materials were processed by extrusion, followed by injection-molding processes. The mechanical, thermal, microstructural, and surface properties were assessed. While the addition of untreated wool fibers to the plasticized PLA/MLO matrix caused a general decrease in the mechanical properties, the TVS treatment was able to slightly compensate for such mechanical losses. Additionally, a shift in cold crystallization and a decrease in the degree of crystallization were observed due to the fiber silane modification. The microstructural analysis confirmed enhanced interaction between silane-modified fibers and the polymeric matrix. The inclusion of the fiber into the PLA/MLO matrix made the obtained material more hydrophobic, while the yellowish color of the material increased with the fiber content.

## 1. Introduction

One of the most promising biobased and biodegradable polymers to replace petroleum-based polymers is poly(lactic acid) (PLA), which is a commercially available biopolyester at a competitive cost that is easily processed using the same technology currently used for traditional plastics [1,2]. Its popularity is mainly due to its renewable origin, inherent biocompatibility, biodegradability, and recyclability as well as its relatively low cost compared to other biodegradable polymers [1,3,4,5]. Firstly, PLA has been widely used in the biomedical sector due to the development of bio-absorbable surgical implant materials, drug delivery systems, or porous scaffolds [2,4]. Then, it was extended to other industrial sectors. For instance, due to its high transparency and easy processing characteristics, PLA is commonly used to produce short-term food packaging products [1,6,7] and agricultural mulch films [8,9]. In addition, the construction and automotive industries have been interested in the use of natural fiber-reinforced plastics and composites, since these fibers are lighter than glass and petrochemical fibers as well as economically and ecologically acceptable [10]. Thus, composites based on PLA are an environmentally friendly option that contribute to a circular economy approach [11]. In fact, they are composed of a biobased and biodegradable polymeric matrix, which can contribute to a reduction in the use of traditional synthetic composites in several fields [12].

Nevertheless, the spectrum of applications of PLA-based materials at the industrial level is still reduced due to its relatively low thermal stability, high brittleness, as well as low elongation at break [1,13,14]. Therefore, several strategies have been used to improve PLA’s performance and increase its industrial exploitation, such as copolymerization [15,16] and blending approaches [1,14,17,18,19]. Blending strategies are preferred, since they are easy and cost-effective for production at the industrial level [19,20]. In this sense, from an industrial point of view, PLA plasticization is very interesting not only to increase PLA’s flexibility but also to facilitate its processability [20,21,22]. Thus, PLA has been plasticized with several plasticizers, including polyethylene glycol (PEG) [23,24], oligomeric lactic acid (OLA) [21,25], and citrate esters (i.e., acetyl tributyl citrate (ATBC) [7,8,22], triethyl citrate (TEC) [20]), etc. Seed oils have gained interest as natural plasticizers in the last few decades [9,14,26]. In this sense, maleinized linseed oil (MLO) has been already studied as an additive of PLA, playing an interesting plasticizing role that helps to overcome PLA’s disadvantages without removing the bio-based character of the final material [7,13,14]. Moreover, MLO has shown its effectiveness, providing a compatibility effect between PLA and other bio-based materials or reinforcements, such as thermoplastic starch or wool fibers [14,27,28].

Another interesting approach to increase PLA’s performance is the development of PLA-based composites and nanocomposites. Due to the growing environmental concern, composite manufacturing with natural fibers as well as biodegradable polymers instead of synthetics allows decreasing the carbon footprint [29]. However, the use of natural fibers as a reinforcement frequently is not competitive compared to the properties provided by synthetic fibers. Nevertheless, natural reinforcements are advantageous as they comes from renewable sources, are biodegradable, and have lower costs [7,30]. Thus, natural fiber-reinforced composites are more and more often considered in a wide range of industrial sectors (i.e., as regenerative medicine [31]; for interior and outdoor products [32]; in the construction, automobile, aerospace industries [33,34]). Moreover, the possibility of profiting from the natural fibers obtained from agri-food industries’ wastes is gaining considerable interest [27,28,34,35]. In this sense, wool fiber by-products are a rich keratin-based natural resource waste that is not subject to adequate forms of valorization [36]. Wool fibers are stable in the temperature range habitually used for thermoplastic material processing and, thus, they have been proposed as a keratin-based reinforcing additive for several polymeric matrices [22,27,28]. A huge amount of crude fibers and short fibers are abandoned in the rangeland during shearing every year, and the use of these wool residues which are chemically identical to longer fibers allows reducing the number of wool by-products from the textile industry with correct waste [28,31,36]. Even more, wool represents an important by-product from the sheep milk and cheese industry, since the wool derived from dairy sheep is not the most wanted for the textile industry [27]. Thus, it would be positive to increase the use of dairy sheep wool fiber from sheep milk production as a useful material reinforcement.

Although natural fibers present plenty of advantages for the development of polymer composites, the frequently lower mechanical properties of the obtained materials often restrict the use of these materials in the industrial sector [27,28]. Thus, the surface modification of the natural fiber is highly suggested to improve the chemical fiber/matrix interaction and the roughness of the fiber, which consequently will increase the adhesion between the components of the final composite [30,33]. In this regard, an increase in adhesion has been observed between PLA-based composites reinforced with cellulosic reinforcements after the use of an epoxide-based reactive agent [37]. In addition, on PLA systems reinforced with wool fibers, some silane and alkoxide-based coupling agents have been studied to improve the compatibility and increase the interaction between the fiber/matrix in the final composites [28]. In fact, the most promising strategies to increase the compatibility between natural fibers and biopolymeric matrices are polymer grafting, the silane coupling of fibers, or the addition of compatibilizer agents [31,37,38]. Although surface modification is commonly applied for inorganic reinforcements, such as silica [39,40] or titanium dioxide [41], the silane treatment can also be successfully applied for organic components or natural fibers, such as cellulose or wool [28,42]. Moreover, silane treatment fiber functionalization is preferred for the industrial sector with respect to other functionalization strategies (i.e., grafting reaction) due to its relative simplicity. Thus, to overcome the limited compatibility of natural fibers and polymers, silane treatment is commonly applied to natural fibers intended for the development of composites due to its high efficiency, easy processing, and favorable results [36,43]. The silane coupling reaction and its mechanism have been widely described for various natural fibers and can be found elsewhere [42,44]. Among the favorable characteristics reported, it is possible to find good thermal, mechanical, and physical properties, with the high strength performance of the final silane-treated composites standing out. Therefore, the surface modification allows increasing the compatibility of the reinforcement of the system by inducing a favorable organic/inorganic interaction at the reinforcement surface, which prevents agglomeration [39,41].

However, from a technological point of view the use of bio-origin fibers and their further modification requires some purification steps which increase the cost of the final material and involve additive preparation processes that sometimes become new challenges for the production process [45].

In this regard, the washing conditions of sheep wool fibers were optimized in a previous work, and those fibers were further used for reinforcing plasticized PLA with 10 phr of MLO (PLA/MLO). Increasing amounts of sheep wool fibers were added from 1 to 11 phr to assess the influence of the fiber amounts on the overall mechanical and thermal properties of PLA/MLO [27]. Nevertheless, a low adhesion of sheep wool fibers to the plasticized polymeric matrix was detected, resulting in a decrement in the tensile and flexural properties [27]. Then, to increase the compatibility of sheep wool fibers with the polymeric matrix, wool fibers were surface-functionalized with silane coupling agents to improve the fiber/matrix compatibility [28]. Some studies report that the silanization of wool fibers exerts a remarkable effect on the mechanical and thermal properties of the final composites [28,36,44]. Silane treatment has not been deeply studied yet as strategy to increase the interaction between the PLA/wool fiber. In our previous work, we studied the effect of tris(2-methoxyethoxy)(vinyl) silane (TVS), among other silane-based compatibilizers, as a sheep wool fiber surface modifier in a concentration of 1 phr. The silane-functionalized fibers were used in 1 and 10 phr to reinforce PLA/MLO system. The results showed that the functionalization of sheep wool fibers with 1 phr of TVS increased the tensile and flexural properties as well as increased the cold crystallization temperature [28]. Thus, with the aim to further improve the compatibilization of sheep wool fibers with a PLA/MLO matrix, in the present work a higher concentration of the TVS coupling agent of 2.5 phr is proposed. The effect of the coupling agent concentration was studied at two concentrations, 1 and 2.5 phr, with respect to the sheep wool fiber. Meanwhile, for the development of composites both silane-functionalized sheep wool fibers were added in 1, 5, and 10 phr to the PLA/MLO matrix to get information about the influence of the fiber amount as well as the silane treatment concentration on the mechanical, thermal, and surface wettability properties of the final green composites intended for injection-molded parts with interest in a wide range of industrial applications (i.e., automotive parts, indoor and outdoor applications).

## 2. Materials and Methods

### 2.1. Materials

Poly(lactic acid) (PLA) Ingeo Biopolymer 6201D was provided by Nature Works LLC (Minnetonka, MN, USA) (2% D-lactic acid, density of 1.24 g/cm^3^, and a melt flow index (MFI) between 15 and 30 g (10 min, 210 °C)). Maleinized linseed oil (MLO) was used as a plasticizer, commercial-grade Veomer Lin was supplied by Vendeputte (Mouscron, Belgium) (viscosity of 10 dPas at 20 °C and an acid value of 105–130 mg KOH/g). Wool was obtained in the form of raw material from a cheese industry from Basque Country (Spain), which was cleaned beforehand and prepared following a procedure that was previously reported [27]. In brief, the fibers were cut and cleaned using a water solution of 0.2 %wt. sodium carbonate and 2 %wt. detergent. Then, they were carded to obtain oriented fibers in one direction and cut into 1 cm lengths, achieving a ratio L/d (length/diameter) of 200. As a coupling agent, tris(2-methoxyethoxy)(vinyl) silane (TVS) was used (density of 1.034 g/mL at 25 °C and refractive index n_20_/D of 1.43) and was supplied by Sigma Aldrich (Schnelldorf, Germany). The chemical structures of the components employed are shown in Figure 1.

### 2.2. Methods

#### 2.2.1. Silane Wool Fiber Treatment

A schematic representation of wood fiber silane treatment is shown in Figure 2.

Wool fibers were modified using 1 phr and 2.5 phr of TVS coupling agent per 100 parts of wool fiber. First, the TVS was added to acetone 99% solution, heated up to 40 °C, and mixed for 30 min. Then, the wool was immersed in the coupling agent solutions and stirred for 2 h at 1000 rpm and further dried in an air-circulating oven at 80 °C for 48 h until complete solvent evaporation. Untreated wool was also used as a reference.

The coupling reaction between the TVS and wool is shown on the upper side of Figure 2. Firstly, the hydrolysis of the coupling agent takes place in an acetonic solution, where water allows the hydrolysis process and acetone has the role of the solvent. Then, wool fibers with adsorbed water on their surface tend to create hydrogen bonds with TVS hydroxyl groups. Finally, the heat treatment allows water evaporation, leading to the grafting of the silane molecule on the wool fiber surface [36,42,44]. To verify the correct silane surface treatment of the wool, unmodified wool fibers, the two silane-treated wool fibers modified with 1 phr and 2.5 phr of TVS coupling agent (labeled UW, W-1TVS, and W-2.5TVS, respectively), and the pure TVS were analyzed by an Attenuated Total Reflectance Fourier-Transformed Infrared Spectroscopy (ATR-FTIR) analyzer (Perkin-Elmer Spectrum BX, Beaconsfield, UK). All the samples were examined from 4000 to 600 cm^−1^, using 20 scans and 16 cm^−1^ of resolution.

The thermal stability of the wool fibers was studied by thermogravimetric analysis (TGA) in Linseis TGA PT1000 (Selb, Germany) conducted under isothermal mode at 180 °C for 30 min under air conditions.

The X-ray diffraction (XRD) patterns of the unfunctionalized and silane-treated fibers were obtained by BRUKER AXS D5005 equipment (SCSIE Universitat de València). The patterns were obtained from a diffractometer using Cu Kα radiation with a scanning step of 0.02° between 2.5° and 40° in 2θ at 40 kV, with a collection time of 10 s per step. Then, the obtained raw curves were smoothed using the Savitzky–Golay method with 25 points of window in the OriginPro software.

Fibers were also observed by scanning electron microscope (SEM) in a Phenon SEM electron FEI microscope (Eindhoven, the Netherlands) using a voltage of 5 kV. The wool samples were previously coated with a gold-palladium alloy in a Sputter Coater Emitech SC7620, Quorum Technologies (East Sussex, UK) to allow conductivity.

#### 2.2.2. Composite Manufacturing

PLA was plasticized with 10 phr of MLO with respect to neat PLA based on previous works [14,27,28]. The plasticized PLA/MLO samples were further loaded with wool fibers for the development of composites. The PLA/MLO matrix was filled containing 1, 5, and 10 phr (part of fiber per 100 parts of neat PLA) of wool fiber as well as silane-treated wool fibers with TVS. Unmodified wool composites were also prepared and assessed as comparative materials. Thus, eleven formulations were prepared and named, as shown in Table 1.

All the components of the formulations were previously dried and then pre-mixed in a plastic container. Then, the materials were extruded in a Dupra S.L (Castalla, Spain) co-rotating twin-screw extruder, L/D ratio of 25, using a temperature profile of 150-170-180-180 °C (from hopper to die) with a rotation speed 10 rpm. The processing temperature profile was chosen to assure the melt of the PLA as well as to avoid wool fiber degradation, which can occur at above 200 °C [28,46]. After this, each formulation was ground in a Silmisa mill (Onil, Spain) to obtain pellets between 3.5 and 4 mm. Then, the materials were extruded and milled again at the same processing conditions to assure a better interaction among the components in the final compounding. Finally, the materials were injection-molded in a Erinca Sprinter-11, injection-molding machine (Barcelona, Spain) to obtain standard test samples, using a processing temperature profile of 160-170-180-180 °C (from hopper to die). The injected specimens were used for the further characterization of each composite formulations.

#### 2.2.3. Composites Characterization

##### Mechanical Properties Assessment

Tensile assessments were performed in an Ibertest ELIB-50-W universal test machine (Madrid, Spain), with a load cell of 5 kN and crosshead rates of 10 mm/min. The standard test method ISO 527 was employed [47]. Charpy’s impact resistance was assessed in a Metrotec (San Sebastian, Spain) equipment using a 6 J pendulum under the ISO 179 standard [48]. A hardness test was performed using a Shore D durometer model 673-D (Instruments J. Bot S.A., Barcelona, Spain), under ISO 868 [49]. In each test, five specimens were characterized, and the mean and standard deviation of the measurements are reported.

Significance in the data differences among the formulations was statistically analyzed with the OriginPro 8 software (OriginLab, Northampton, MA, USA) using a one-way analysis of variance (ANOVA) at 95% confidence level according to Tukey’s test.

##### Thermal Characterization

Differential scanning calorimetry (DSC) was performed in a Mettler DSC 821e (Toledo, Spain) calorimeter. The temperature cycles consisted of a first heating from 30 to 180 °C, followed by a cooling cycle from 180 to 30 °C. Finally, a second heating from 30 to 220 °C was conducted. All the cycles were conducted under a nitrogen atmosphere using a heating rate of 10 °C/min with a flow of 30 cm^3^/min. The glass transition temperature (T_g_), melting temperature (T_m_), and cold crystallization temperature (T_cc_) were determined from the second heating of the DSC curve to erase the thermal history. The degree of crystallinity (X_c_%) was calculated according to Equation (1):(1)$$X_{c}\left(\%\right)=\frac{\Delta H_{m}-\Delta H_{CC}}{f\times \Delta H_{m}^{c}}\times 100,$$
where $\Delta H_{m}$ (J/g) is the melting enthalpy and $\Delta H_{CC}$ (J/g) is the cold crystallization enthalpy of the material. The $\Delta H_{m}^{c}$ (J/g) is the calculated melting enthalpy of purely crystalline PLA, at 93 J/g [50], and f is the weigth fraction of PLA in the formulation.

Thermogravimetric Analysis (TGA) was carried out in a Linseis TGA 1000 (Selb, Germany). The thermal curve was assessed from 30 °C to 700 °C at a heating rate of 10 °C/min under a nitrogen atmosphere using a flow of 30 cm^3^/min. The onset degradation temperature (T_5_) was calculated at 5% of mass loss. The temperatures of the maximum decomposition rate (T_max_) were calculated as the temperature of the peak from the first derivative of the TGA curves (DTG). The mass losses at 300 and 350 °C are also reported.

In the DSC and TGA tests, three samples of each formulation were assessed, and the average values and the standard deviations are reported. Significant differences in the properties were statistically analyzed under the same parameters reported above.

##### X-ray Diffraction

The crystalline phases of the bionanocomposites were examined by X-ray diffraction (XRD) equipment (BRUKER AXS D5005, SCSIE Universitat de València). Scanning was performed on square bionanocomposite film surfaces (15 mm × 15 mm) mounted in an appropriate sample holder. The patterns were obtained from a diffractometer using Cu Kα radiation, with a scanning step of 0.02° between 2.5° and 40° in 2θ with a collection time of 10 s per step, while the voltage was held at 40 kV. The obtained raw curves were smoothed using the Savitzky–Golay method with 25 points of window in the OriginPro software.

##### Composites Surface Properties Evaluation

A microscopic analysis of the fracture surface of the impact specimens was performed by SEM (Phenon SEM electron FEI microscope, Eindhoven, the Netherlands) using a voltage of 5 kV. The fractured samples were previously coated with a gold-palladium alloy (Sputter Coater Emitech SC7620, Quorum Technologies, East Sussex, UK) to allow conductivity.

The wettability was determined using an EasyDrop-FM140 optical goniometer from Kruss Equipments (Hamburg, Germany) equipped with a camera and Drop Shape Analysis software based on the determination of the static contact angle (WCA—theta angle) of a distilled water drop created on a surface of each sample; after 30 s, the WCA of each drop was measured at room temperature. The average values of five measurements were reported with a maximum standard deviation of 3% [51].

The color measurements of the samples were performed in a Colorflex-Diff2 458/08 colorimeter from HunterLab (Reston, VA, USA) using the CIELab space—that is, by measuring and comparing the L, a*, and b* coordinates, which are concerned with the change in lightness of the color (L), the intensity of the green and red colors (a*), and the intensity of the blue and yellow colors (b*).

For the wettability and color evaluation, 10 measurements were obtained and the means and standard deviations of the values are reported. The significant differences were statistically analyzed using the same parameters previously described.

## 3. Results and Discussion

### 3.1. Silane Wool Fiber Treatment

The FTIR spectra obtained for unmodified wool and modified wool fibers with 1 and 2.5 phr of TVS are reported in Figure 3. In addition, TVS spectrum is reported for comparison. Comparing the treated (W-1TVS and W-2.5TVS) and untreated wool (UW) samples, no significant reduction in the –OH group peak at 3300 cm^−1^ in the wool fibers is observed. Nevertheless, some changes in the TVS-modified wool fibers can be observed in the region between 2880 and 2980 cm^−1^, where the peaks after modification present a slightly higher intensity. Those peaks have been associated with –CH_3_ bonds [28] and are also present in the TVS FTIR spectrum. However, the highest increase in the peaks’ intensity in that region was observed for wool treated with 1 phr of TVS. This behavior agrees with the results reported in PLA/MLO systems using wool fibers modified with several coupling agents, where silane treatment causes a very slight change in the intensity of this peak [28]. A positive interaction between the wool fibers and TVS can be clearly observed in the region between 700 and 1300 cm^−1^, where the peaks are associated with the presence of Si–O–Si and Si–O–R bonds. The most significant change in peak intensity has been observed for wool treated with 2.5 phr of TVS, when compared to the peaks at 1084 and 763 cm^−1^, showing an effective silane coupling to the wool fiber surface. The absorption band at 763 cm^−1^ corresponds to the –Si–C– symmetric stretching bond [52], and as it was shifted to lower wavenumber values (760 cm^−1^) in the silane-treated fiber with 2.5 phr of TVS, it may indicate a coupling effect. Moreover, there is a displacement in the asymmetric stretching of Si–O–Si from 1084 to 1120 cm^−1^ in silane-treated fibers, which suggests a positive interaction between the TVS simple structure and the TVS linkend on the wool fiber surface, confirming the success of the fiber modification treatment [40]. Additionally, on the neat TVS spectrum peaks at 1023 and 840 cm^−1^ could be observed which can be associated with the stretching vibration of the Si-O–C bond and the stretching vibration of alkoxy groups. Similar bonds were observed for vinyltriallyloxysilane monomer used as a coupling agent and disappeared after a successful coupling reaction [53]. Furthermore, the lack of mentioned peaks on a spectrum of the W-2.5TVS sample indicates no presence of unreacted coupling agent.

The thermal stability of sheep wool fiber as well as TVS-treated wool fibers was studied at the composite processing temperature (180 °C) and is shown in Figure 4a. As expected, the functionalized wool fibers showed a higher thermal stability at the processing temperature than the untreated wool fibers, but have a common thermal behavior when degraded in air atmosphere. The untreated fibers showed the same thermal stability as silane-treated ones for times lower than 3.5 min, losing around 5% of their initial weight, which corresponds to moisture evaporation and the volatilization of low molecular weight compounds. Then, the sheep wool fiber components (i.e., keratin) start the decomposition. The comparison of silane-treated fibers evidenced that sheep wool fibers with a higher content of TVS (2.5 phr) have an increased thermal stability with respect to wool functionalized with 1 phr of TVS.

The XRD patterns of the sheep wool fibers (Figure 4b) show two peaks of keratin crystalline structure responsible for the α-helix structure at around 9.2° and for the β-sheet structure at around 20.9° [54]. The broad character of the peaks obtained during the measurements indicates also the presence of an amorphous structure of wool. However, the modified fibers (W-1TVS and W-2.5TVS) do not show the presence of an additional crystalline structure, indicating that the coupling agent attached to the fiber is in an amorphous form [55]. Additionally, the XRD patterns showed that modification procedure does not affect the crystalline structure of wool, as no peak broadening, explaining any faults or defects in keratin, was observed.

The functionalized microstructure of sheep wool fibers was observed by SEM. While, in unmodified fiber, cuticle cells are adhered to the fiber bulk (Figure 4a), in the silane-treated fibers there is an increased detachment of cuticle cells from the fiber bulk with increasing silane treatment concentrations (Figure 5b,c). It is known that the cell morphological structure of wool fibers is characterized by the presence of three main components, including the cuticle, which is a thin layer surrounding the fiber bulk (cortex) via the cell membrane complex, occasionally referred to as the intracellular cement. The increased detachment of cuticle cells from the fiber bulk has been related to a successful silane treatment in which its selectivity reacts with the cell membrane complex, while the hydrophobic external layer surrounding each cuticle cell swells [36].

### 3.2. Composite Charactarization

#### 3.2.1. Mechanical Properties

The tensile mechanical properties of the PLA/MLO-wool composites are shown in Figure 5. As expected, the tensile strength (Figure 6a) and Young’s modulus (Figure 6b) of PLA decreased with the addition of MLO. Meanwhile, the elongation at break increased, confirming the plasticizing effect of MLO, as was already reported in the literature [14,27].

The tensile strength of the composites reinforced with 1 phr of both untreated and treated fibers significantly decreased (*p* < 0.05) with the increasing amount of wool. This behavior is indicative of weak interaction between the PLA/MLO matrix and the untreated wool [46]. Additionally, no significant differences (*p* > 0.05) between the treated and untreated wool composites were detected.

Concerning the Young’s modulus values, the addition of 1 or 5 phr of unmodified and modified wool fibers with 1 phr of TVS mainly maintained the Young’s modulus of PLA/MLO. Meanwhile, the silane-treated wool fibers with 2.5 phr of TVS produced a significant increment in the Young’s modulus, showing the positive interaction between the better surface-modified wood fibers with the plasticized polymeric matrix. Higher amounts of wool fibers of 10 phr mainly maintained the Young’s modulus of PLA/MLO. The negative effect on the modulus when increasing the content of natural fillers in PLA-based composites has been also described when using not-compatibilized lignocellulosic fillers [4,27,28,56].

Finally, as expected the elongation at break values of the plasticized PLA/MLO decreased with an increasing amount of wool fibers. Nevertheless, it should be highlighted that while untreated wool fibers lead to composites with a somewhat higher flexibility than neat PLA, the TVS treatment leads to an improvement in the ductility of the composites. A clear positive effect on the elongation at break of the silane treatment applied to the wool fiber is observed in those composites loaded with 1 and 5 phr of fibers treated with 1 phr, particularly in those treated with 2.5 phr of TVS. In this sense, the PLA/MLO-1W-1TVS, PLA/MLO-1W-2.5TVS, and PLA/MLO-5W-2.5TVS formulations (*p* > 0.05) exhibited the highest value of this property among all the studied materials. In fact, the elongation at break has increased in PLA/MLO-1W-1TVS and PLA/MLO-1W-2.5TVS over 60% with respect to the 1 phr untreated wool material (PLA/MLO-1W), while it increased almost 40% in PLA/MLO-5W-2.5TVS with respect to the 5 phr untreated wool material (PLA/MLO-5W). Additionally, in materials containing 10 phr of wool, the elongation at break increased after the TVS treatment, changing from 8.5% in PLA/MLO-10W to 12.4% for PLA/MLO-10W-2.5TVS. These results confirm that the chemical modification of sheep wool fibers improves the interaction between the reinforcing wool and PLA/MLO matrix, especially at lower contents of fibers (1 and 5 phr) and when using the highest amount of TVS of 2.5 phr. However, at a relatively high content of wool (10 phr), the coupling agent treatment is not effective enough because of the high incompatibility of the fibers and the polymer matrix at this concentration [27,46].

In general, the tensile properties trend to increase with the increasing amount of wool content in the composite up to 5 phr, while at 10 phr of sheep wool fibers the overall mechanical properties tend to decrease due to the high incompatibility of sheep wool fibers in the PLA/MLO polymeric matrix. In fact, the mechanical performance depends on the interaction between the polymeric matrix and the fibers, the compatibility and adhesion among them, and the amount of fiber in the final composite [28,56]. However, based on the statistical analysis, the modification of wool fibers with TVS is avoiding this loss of properties, especially detected at high wool and TVS coupling agent contents, because of the better interaction between the plasticized PLA matrix and the modified surface of sheep wool fibers.

Concerning to the Charpy’s impact energy, reported in Figure 6, the impact resistance is decreasing with increasing content of unmodified wool. The decrease in the property can be explained by the fiber breakage and delamination of the material, resulting in lower impact resistance for the higher content of fiber in the composite [33]. Comparing the lowest and the highest wool concentrations (1 and 10 phr, respectively), it is possible to observe that the modification with TVS of the wool fiber tends to increase the impact resistance, even though there are no statistically significant differences between the wool-containing formulations. In the case of high amounts of wool, the treatment with 2.5 phr of TVS showed a stronger interaction than 1 phr of TVS or the untreated wool. The same tendency has been observed for composite laminates, where the presence of the coupling agent allowed an increase in the load transfer between the fiber and polymer matrix [29]. Additionally, even when PLA/MLO-1W-1TVS exhibits the highest Charpy’s impact energy of all the studied formulations (14.9 kJ/m^2^), the statistical analysis shows that there are no significant differences (*p* > 0.05) between the formulations. The lowest value of this property is exhibited by PLA/MLO-10W-1TVS. According to literature, in general PLA composites with natural fibers tend to increase the impact strength of the material.

Finally, referring to the hardness (Figure 7), it is possible to observe that both untreated and treated wood fibers increase the hardness of neat PLA and PLA/MLO formulations, showing the positive reinforcing effect of wool fibers on the PLA matrix. However, the Shore D values tend to decrease with the increasing concentration of untreated wool. If the untreated wool materials are compared, significant differences (*p* < 0.05) between them can be observed. The increase in the wool content from 1 to 10 phr of unmodified wool changes the hardness from 78.5 to 76.9, respectively. Hardness loss has been already observed in PLA reinforced with untreated wool fibers [27]. However, the surface modification of the wool fiber tends to slightly increase the hardness values, even though they are not significantly different (*p* > 0.05), if the same content of treated and untreated wool fiber materials is compared.

Summing up the mechanical properties, an increase in the untreated wool fiber concentration tends to decrease the tensile properties of the PLA/MLO based composites. Nevertheless, the use of TVS coupling agent increases the Young’s modulus and the elongation at break, especially in the formulations containing 1 and 5 phr of wool content, as well as 1 and 2.5 phr of TVS treatment in the fibers, confirming the success of the sheep wool fiber functionalization.

#### 3.2.2. XRD Diffraction Pattern

The XRD patterns (Figure 8) present a single broad peak occurring at around a 16.4° assigned PLA crystalline structure to phase α, and confirm that PLA had no polymorphic transition. The peak intensifies for the PLA-MLO formulation can be explained by the increased PLA chain mobility and the better forming of the crystalline structure of PLA [57,58,59]. The modification of the PLA-MLO system with wool fibers shows the comparable forming of the crystalline structure of neat PLA films, as filler decreases the mobility of polymer chains. Wool-containing formulations do not show additional peaks related to the wool crystalline structure, which can be caused by low concentrations of fiber.

#### 3.2.3. Thermal Properties

The thermal parameters extracted from the DSC and TGA assessments are shown in Table 2. Figure 9 shows the second heating DSC curves of all the studied materials. The plasticization of the PLA matrix with MLO was confirmed with a slight reduction in the T_g_ values in all the plasticized formulations, compared with the neat PLA. Ferri et al. already studied PLA plasticized with different amounts of MLO (from 5 phr to 20 phr) and observed that, over 5 phr of MLO, a small reduction in the T_g_ values of PLA can be detected due to a saturation effect. However, higher amounts of MLO of 10 phr are required to increase the flexibility of these materials [14]. The further incorporation of untreated and treated wool fiber did not significantly (*p* > 0.05) modify the T_g_ values.

The PLA T_cc_ decreased with the MLO incorporation, as previously observed for frequently used PLA plasticizers such as ATBC [8], as well as for already-reported PLA plasticized with MLO [14]. The incorporation of both untreated and treated wood fibers leads to an increase in the T_cc_, showing the ability to affect the crystallization of PLA increasing the crystal nuclei, which requires more thermal energy to re-crystallize PLA. The increasing wool content in the PLA/MLO matrix tends to shift the cold crystallization to higher values (as shown in Figure 9 and reported in Table 2). Even though most of the T_cc_ values are statistically equivalent (*p* > 0.5), the materials with 10 phr of modified wool tend to increase their cold crystallization temperature. PLA/MLO-10W-1TVS and PLA/MLO-10W-2.5TVS show a higher value of T_cc_ (about 107 °C), while the formulation that contains 1 phr of untreated wool presents the lowest cold crystallization temperature (T_cc_ for PLA/MLO-1W = 103.1 °C). A similar shift in the cold crystallization temperature was also observed for PLA reinforced with different silane-modified wool fibers [28]. Additionally, a similar synergic effect on the cold crystallization temperature of PLA was observed in previously reported works on plasticized PLA-based blends loaded with lignocellulosic fillers [7,8]. Referring to the coupling agent, the increase in concentration does not significantly change the T_cc_, as is reported in Table 2 and shown in Figure 9b.

Regarding the melting temperature, neat PLA showed a slight change in the baseline of DSC as an exothermic peak before melting (Figure 9a), suggesting that it crystallizes and melts upon heating. This behavior was also observed in the PLA/MLO sample, but this small crystallization peak decreased with an increasing amount of fiber. In fact, it practically disappeared in PLA/MLO-10W (Figure 9a). Meanwhile, no baseline changes were observed prior to the melting in the case of functionalized silane fiber-based composites (Figure 9b), confirming the absence of different crystals. No significant difference (*p* > 0.05) was observed in the melting temperature (T_m_) of composites compared to neat PLA. A higher crystallinity degree was observed for plasticized PLA/MLO formulation, in accordance with the XRD results. The untreated wool-based composites showed higher crystallinity degrees (X_c_%) than neat PLA (Table 2). Similar behavior was observed by Islam et al. for oil palm empty fruit bunch, where PLA and natural fiber have the ability to establish hydrogen/covalent bonds after melt processing [60]. According to Moshiul Alam et al., also alkali, ultrasound, and combined treatments allow increasing crystallinity degrees (X_c_%) of PLA-based formulations [61]. On the contrary, the further silane surface modification of wool fibers with TVS causes the crystallinity degrees to remain equal to or lower than the degree for the neat PLA, depending on the wool content. Such behavior may be explained by limited hydrogen bonding interactions between sheep wool fiber and PLA after processing.

The thermal stability of the formulated materials has been assessed by means of TGA, and the results are shown in Table 2 and Figure 10. The beginning of the degradation of the material has been evaluated by comparing the onset degradation temperatures, determined at 5% of the mass loss of each composite. In all the plasticized PLA/MLO formulations, a decrease in the T_5_ was observed, as frequently occurs for PLA plasticized formulations [7,14,28]. The highest T_5_ was observed for PLA modified with 1 phr of untreated wool (T_5_ = 326.3 °C), making the PLA/MLO-1W the most thermally stable among the studied formulations. Increasing the wool content prone to significantly decrease the thermal stability of the composites, which can be explained by the relatively low thermal stability of sheep wool fiber. The surface modification of wool fibers has a significant influence on formulations containing 10 phr of wool, where the T_5_ is found to be lower than 300 °C. As reported for flax/glass fiber composites, the surface treatment allows the natural fiber to decompose over 300 °C, improving their thermal stability [33]. However, such an improvement has been observed only in composites reinforced with 1 and 5 phr of wool fibers. Finally, the T_5_ exceeds 280 °C for all the composites, confirming that the materials are stable in the processing conditions used here and no degradation of natural additives takes place during processing. The mass loss occurring from 30 up to 250 °C (shown in Figure 10a,b), is related to the beginning of wool fiber degradation, and refers mainly to the evaporation of the humidity in the samples, particularly in unmodified sheep wool fibers and in those formulations with a higher fiber content (PLA/MLO-5W and PLA/MLO-10W, Figure 10a). In general, the T_max_ of neat PLA and all formulated materials is lower than the T_max_ of the PLA/MLO, with the exception of PLA/MLO-5W-1TVS. In fact, the MLO was able to increase the T_max_ of neat PLA, showing a good interaction with the polymeric matrix.

The incorporation of low amounts of sheep wool fibers (1 and 5 phr) mainly maintained the T_max_ of PLA/MLO, but somewhat showed a reduction, having no significant differences from the neat PLA sample. The silane treatment in 1 phr-reinforced materials (PLA/MLO-1W-1TVS and PLA/MLO-1W-2.5TVS) slightly decreased the T_max_ values, but without showing significant differences (*p* > 0.05). Different behavior was observed in the 5 phr-reinforced materials, in which the silane treatment at 1 phr of TVS—that is, the PLA/MLO-5W-1TVS formulation—was able to increase the T_max_ of its not-functionalized counterpart (PLA/MLO-5W), and also showed a somewhat higher value than PLA/MLO (*p* > 0.05), suggesting that TVS was able to increase the compatibility of the fibers with the plasticized PLA/MLO matrix. The incorporation of the highest amount of fibers decreased the T_max_ value of PLA/MLO, reaching similar values to that of neat PLA (*p* > 0.05). Nevertheless, the silane treatment of 10 phr wool fibers at 2.5 of TVS showed no significant values with PLA/MLO, suggesting that the functionalization countered the negative effect produced by the high amounts of fibers, enhancing their compatibility. These results show that the composites developed here are stable to the processing thermal cycles, and the incorporation of the untreated or TVS-treated wool fibers does not largely affect the thermal degradation of the PLA/MLO matrix.

The mass loss was determined at 300 and 350 °C to assess further changes in material degradation. Similarly, to 5% mass loss, at both temperatures (300 °C and 350 °C) lowest mass loss was observed for formulation PLA/MLO-1W. Referring to the mass loss values determined at 300 °C, an increase in the concentration of unmodified wool caused a significant change from 1.2% to 5.8% of mass loss, between the formulations containing 1 and 10 phr of wool fibers, respectively. Although the mass loss at 300 °C is variable for the tested samples, in general the change caused by the further higher wool concentration or surface treatment does not significantly affect the values. However, the highest mass loss values were determined for 10 phr of wool formulations, related to higher wool concentrations. Referring to the mass loss determined for 350 °C, the increase in the wool content in the composites results in a significantly higher mass loss, especially noticeable between all formulations that contain 1 and 10 phr of wool, respectively. The surface silane treatment of the wool fiber with different TVS does not significantly change these values among the formulations with the same wool concentration. Finally, all the tested materials remain stable in the range of material processing temperatures used for both the extrusion and injection process steps.

#### 3.2.4. Composites Surface Characterization

In many applications, polymeric materials are exposed to rapid impact loads (i.e., impact after free-falling, or direct blows or collisions). Thus, since the impact is one of the most sensitive mechanical properties, a SEM study was carried out on the fracture surfaces of the injection-molded pieces after the impact tests to better observe the structural behavior of the developed composites under impact conditions. As the phase of PLA/MLO (not shown) shows a homogenous fracture surface due to the good miscibility between both compounds [14], the SEM analysis was focused on areas showing space between the polymer and fiber and the interaction between them (Figure 11).

The wettability and color parameters of the surfaces of the formulated materials are shown in Table 3. It is known that neat PLA is a moderately hydrophobic material [62], with a static water contact angle of 77°. However, it presents higher wetting properties than traditional plastic such as polypropylene (PP) [63] and low density polyethylene (LDPE) [51], which show static water contact angle around 100°. The incorporation of MLO into the PLA matrix reduced the WCA value; this behavior can be ascribed to the fact that the increased polymer chain mobility in the plasticized matrix increases the diffusion process [7]. The wettability measurements, shown in Table 3, indicate that the modification of PLA/MLO matrix with untreated and TVS-modified wool fibers significantly increases the hydrophobicity of the composites. While the increasing content of untreated wool in the PLA/MLO matrix increases the hydrophobicity of the material, there is not a marked tendency caused by the TVS treatment. It should be mentioned that, in all composites, the surface hydrophobicity increased up to 20% with respect to that of PLA/MLO.

The color changes were measured in the CIELab space, where *L* describes lightness changes, a* indicates the intensity of red or green colors, and b* indicates the intensity of yellow or blue colors. According to the lightness evaluation, the highest value was reported for the PLA/MLO-1W-1TVS formulation (*L* = 51.16), where the darkest formulation was PLA/MLO-10W (*L* = 37.73). Samples modified with higher wool concentrations show a significant decrease in the L value, in accordance with the darker appearance of PLA composites with increasing contents of sheep wool fibers. A similar darker appearance was reported for lignin-modified PLA composites with increasing lignin contents [3]. On the other hand, wool treatment with TVS significantly increases the *L* value. The lightness change caused by TVS treatment is most visible in the PLA/MLO-10W-2.5TVS formulation, where the *L* value increased by about 20% compared to PLA/MLO-10W.

Significant changes in the a* value were observed with increasing the wool content in composites, passing from 0.48 for the PLA/MLO-1W materials to a deviation towards red tonality (positive values of a*) reaching 8.60 in the PLA/MLO-10W materials. In the case of the b* coordinate, it significantly increased with an increasing wool content. All the formulations followed a tend to yellow color, in good agreement with previous works where the PLA/MLO matrix with 1 phr of wool fibers added showed yellowish materials [28].

Finally, the total color differences (∆E) of the formulated materials show a marked difference in color between the PLA/MLO and the composites with different contents of sheep wool fibers. Considering that the change in color is appreciable to the human eye when the ∆E is higher than 2 [64], it can be concluded that all the formulations were significantly different from the PLA/MLO. The increasing content of wool fibers from 1 to 5 phr shows differences in color, while the composites loaded with 5 and 10 phr did not show appreciable changes. From the color studies, it can be concluded that the surface modification of wool fibers with TVS can significantly influence the intensity of the yellow color of the composite material, especially with an increasing content of wool fibers.

## 4. Conclusions

Sheep wool fibers were successfully surface functionalized with TVS silane coupling agent. The increasing amount of TVS from 1 to 2.5 phr revealed an increased thermal stability as well as an increased detachment of cuticle cells from the fiber bulk, as confirmed by SEM observations. The functionalized sheep wool fibers with 1 and 2.5 phr of TVS were used to reinforce the plasticized PLA/MLO matrix at three different loadings (1, 5, and 10 phr). The results show that improvements in the interaction between the plasticized PLA matrix and the sheep wool fiber were achieved due to the surface chemical silane treatment of the fiber. As frequently occurs when using wool fibers as reinforcement, the increasing wool concentration showed a negative influence on the mechanical performance, particularly at the high fiber content of 10 phr. A decrease in the thermal stability was mainly caused by the higher content of the natural fiber, but showed enough thermal stability at the processing temperature. A weak interaction between the PLA/MLO matrix and the untreated wool was mainly observed, while this negative effect was slightly compensated by the TVS fiber surface treatment. In fact, the surface silane-treated wool fibers at the highest concentration (2.5 phr of TVS), at loading amounts of 1 and 5 phr in PLA/MLO matrix, led to composites with the highest Young’s modulus and elongation at break (PLA/MLO-1W-2.5TVS and PLA/MLO-5W-2.5TVS). SEM observations confirmed the positive interaction between the coupling agent-treated wool fiber surfaces and the plasticized PLA matrix in the final composites. Composites tend to be yellowish and appear darker with an increasing content of wool fiber, while such a visible change was not influenced by the wool surface treatment. The presence of sheep wool fibers made the PLA/MLO system substantially more hydrophobic than PLA and PLA/MLO. Thus, the silane-treated sheep wool fiber-based composites hold promise as alternative sustainable materials for several industrial applications, since they have shown improved mechanical and hydrophobicity performance compared to PLA/MLO and offer the opportunity to obtain novel materials through the revalorization of a dairy industry byproduct.

## Figures and Tables

**Figure 1 polymers-12-02523-f001:**
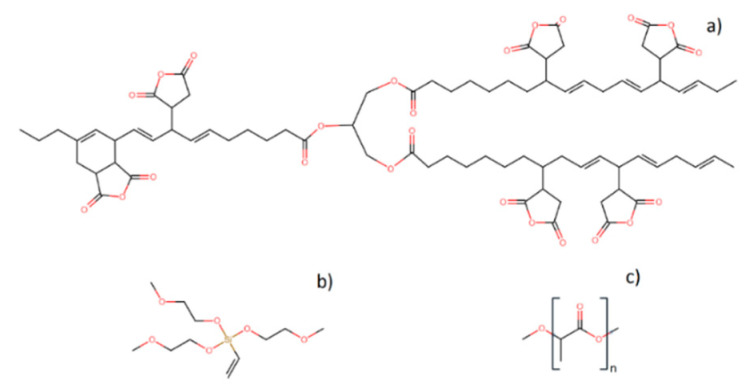
Chemical structure of (**a**) maleinized linseed oil (MLO), (**b**) tris(2-methoxyethoxy)(vinyl) silane (TVS), and (**c**) poly(lactic acid) (PLA).

**Figure 2 polymers-12-02523-f002:**
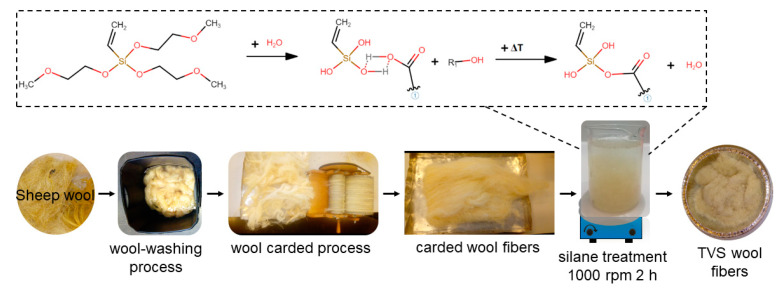
Schematic representation of the sheep wool fiber washing and carding process, the silane treatment with TVS, as well as the coupling reaction between TVS and wool fibers.

**Figure 3 polymers-12-02523-f003:**
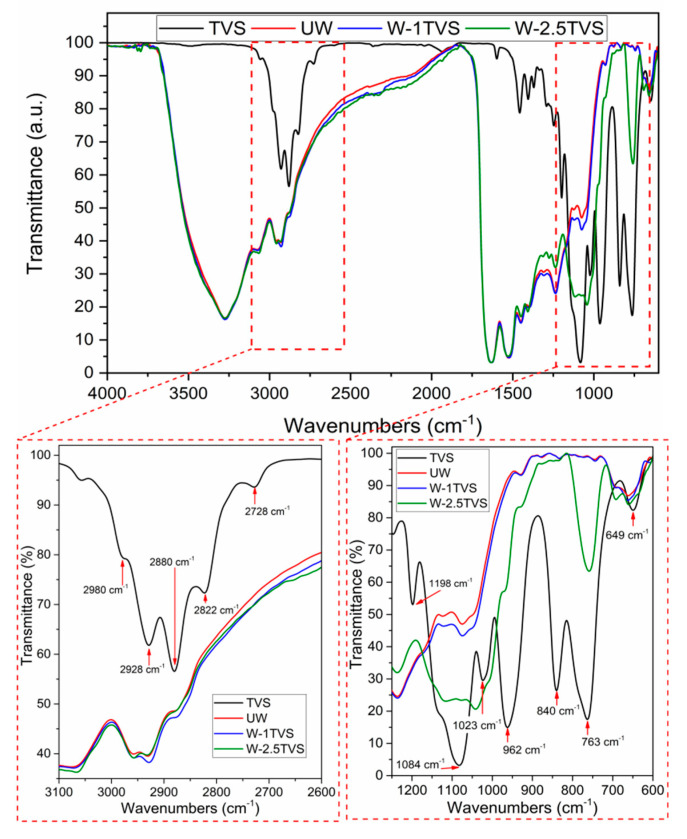
FTIR spectra of TVS and studied untreated wool (UW) and wool treated with different TVS concentrations.

**Figure 4 polymers-12-02523-f004:**
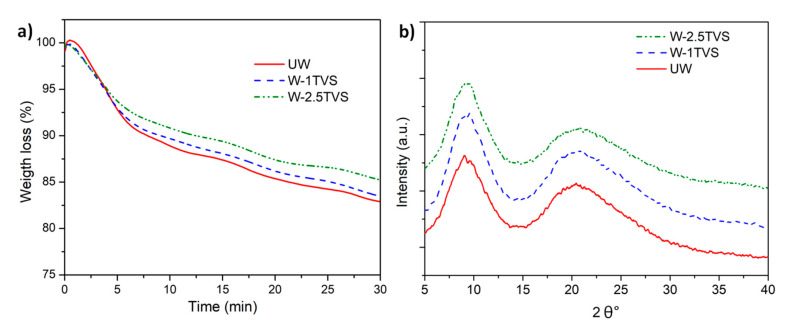
Untreated and silane-treated sheep wool fibers: (**a**) TGA isothermal curves and (**b**) XRD pattern.

**Figure 5 polymers-12-02523-f005:**
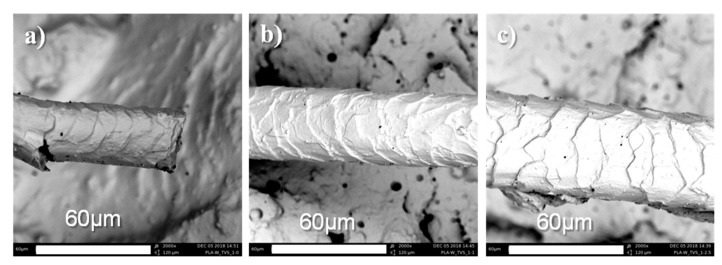
SEM images of (**a**) untreated and silane-treated sheep wool fibers with (**b**) 1 phr of TVS and (**c**) 2.5 phr of TVS.

**Figure 6 polymers-12-02523-f006:**
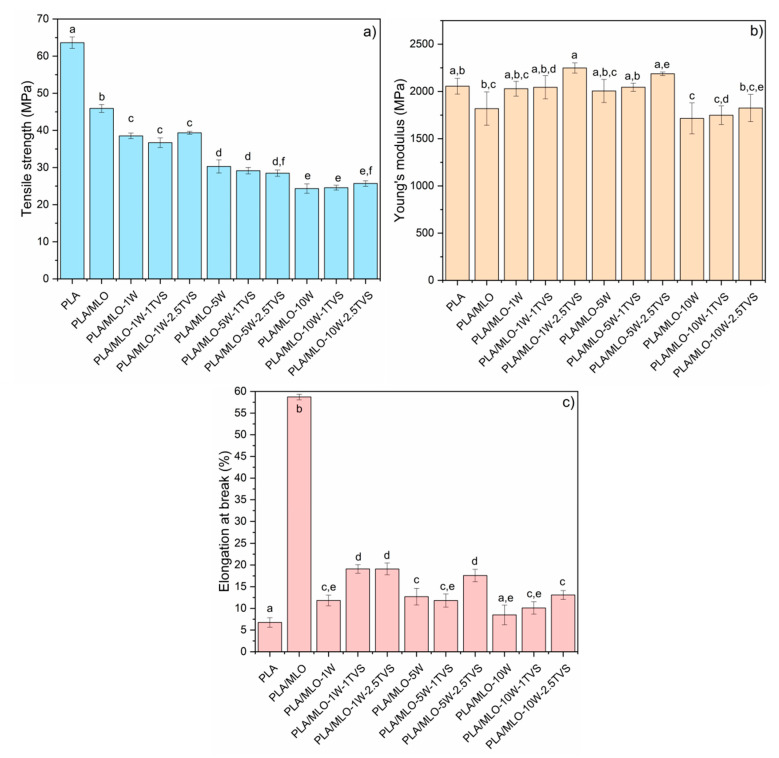
Tensile test results: (**a**) tensile strength, (**b**) Young’s modulus, and (**c**) elongation at break. ^a–f^ Different letters within the same property show statistically significant differences between formulations (*p* < 0.05).

**Figure 7 polymers-12-02523-f007:**
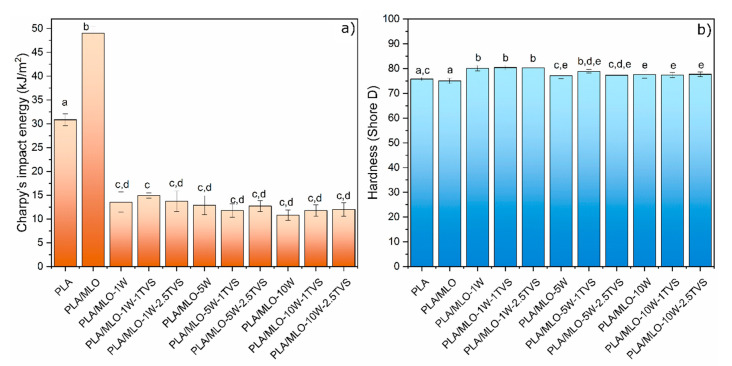
(**a**) Charpy’s impact values and (**b**) Shore D hardness values of the formulated materials. ^a–e^ Different letters within the same property show statistically significant differences between formulations (*p* < 0.05).

**Figure 8 polymers-12-02523-f008:**
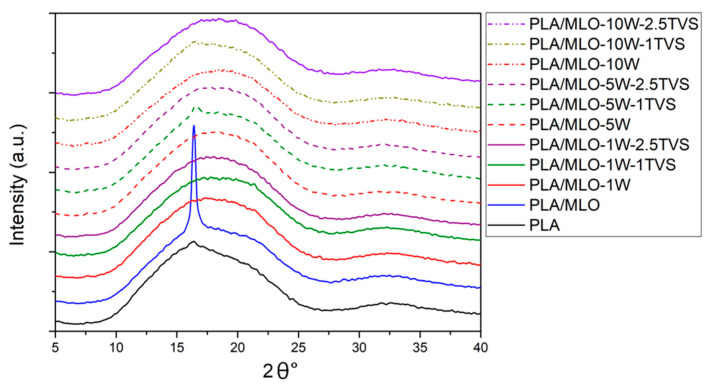
XRD patterns of the formulated materials.

**Figure 9 polymers-12-02523-f009:**
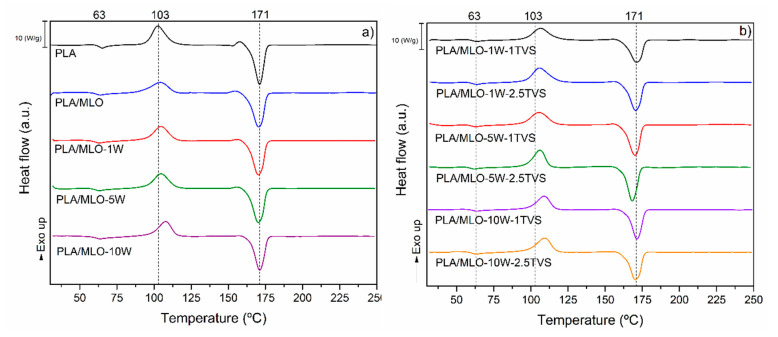
Second heating DSC curves of (**a**) untreated wool composites and (**b**) wool treated with TVS composites.

**Figure 10 polymers-12-02523-f010:**
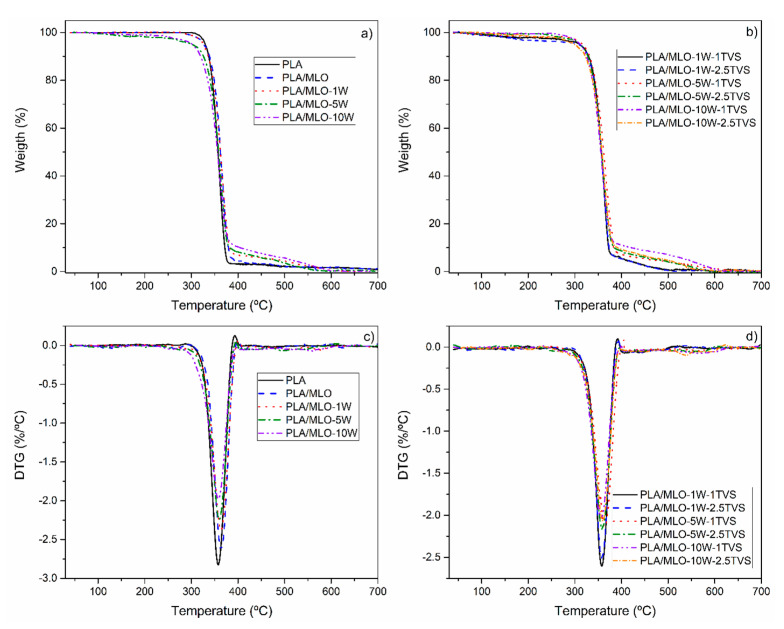
Thermogravimetric (TGA) and derivative thermogravimetric (DTG) curves: (**a**) TGA and (**c**) DTG of PLA, PLA/MLO, and untreated wool composites; (**b**) TGA and (**d**) DTG of TVS wool-treated composites.

**Figure 11 polymers-12-02523-f011:**
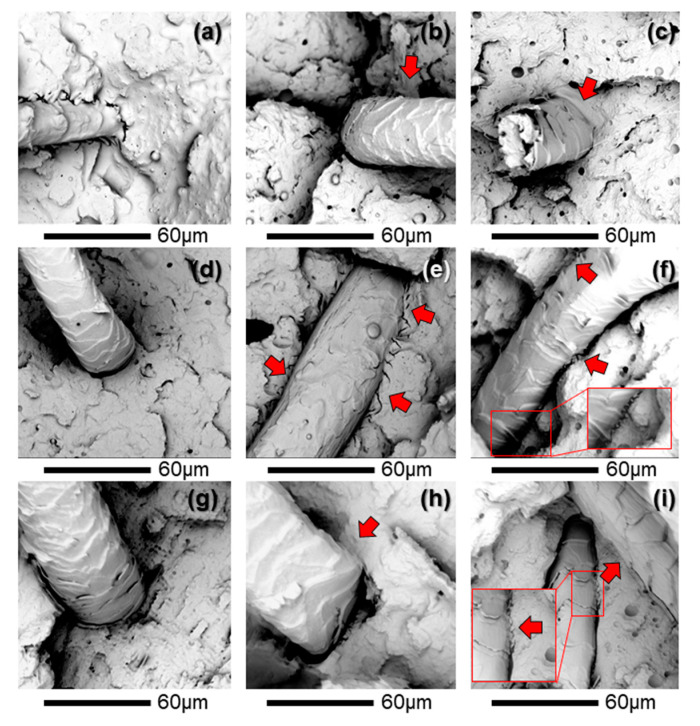
SEM images of (**a**) PLA/MLO-1W, (**b**) PLA/MLO-1W-1TVS, (**c**) PLA/MLO-1W-2.5TVS, (**d**) PLA/MLO-5W, (**e**) PLA/MLO-5W-1TVS, (**f**) PLA/MLO-5W-2.5TVS, (**g**) PLA/MLO-10W, (**h**) PLA/MLO-10W-1TVS, and (**i**) PLA/MLO-10W-2.5TVS.

**Table 1 polymers-12-02523-t001:** Summary of the compositions and labeling of PLA/MLO–wool formulations.

Coding of Formulation	PLA (phr)	MLO (phr)	Wool (phr)	TVS (phr)
PLA	100	-	-	-
PLA/MLO	100	10	-	-
PLA/MLO-1W	100	10	1	-
PLA/MLO-1W-1TVS	100	10	1	1
PLA/MLO-1W-2.5TVS	100	10	1	2.5
PLA/MLO-5W	100	10	5	-
PLA/MLO-5W-1TVS	100	10	5	1
PLA/MLO-5W-2.5TVS	100	10	5	2.5
PLA/MLO-10W	100	10	10	-
PLA/MLO-10W-1TVS	100	10	10	1
PLA/MLO-10W-2.5TVS	100	10	10	2.5

**Table 2 polymers-12-02523-t002:** Thermal properties of the composites.

	DSC				TGA			
Formulation	T_g_ (°C)	T_cc_ (°C)	T_m_ (°C)	X_c_ (%)	T_5%_ (°C)	T_max_ (°C)	Mass Loss at 300 °C (%)	Mass Loss at 350 °C (%)
PLA	63.6 ± 0.6 ^a^	103.3 ± 0.7 ^a^	171.1 ± 0.6 ^a^	16.7 ± 0.2 ^a,d^	331.0 ± 1.0 ^a^	357.3 ± 1.2 ^a,c^	0.05 ± 0.0^a^	21.3 ± 0.9 ^a^
PLA/MLO	60.5 ± 0.6 ^b^	101.8 ± 1.3 ^a^	170.1 ± 0.3 ^a,b^	21.8 ± 0.3 ^b^	326.0 ± 1.0 ^a,c^	362.7 ± 0.6 ^b,d^	1.2 ± 0.1^a,b^	22.9 ± 2.2 ^a^
PLA/MLO-1W	60.2 ± 0.4 ^b^	103.1 ± 0.1 ^a^	168.8 ± 0.3 ^a,b^	18.6 ± 1.1 ^a,e^	326.3 ± 0.6 ^a,c^	361.7 ± 1.6 ^b,c,d,e^	1.2 ± 0.1ª^,b^	22.5 ± 2.2 ^a^
PLA/MLO-1W-1TVS	60.4 ± 0.4 ^b^	106.5 ± 0.1 ^a,b^	170.4 ± 0.9 ^a,b^	13.1 ± 0.6 ^c^	313.0 ± 3.0 ^b,c^	358.3 ± 1.5 ^a,c^	3.3 ± 0.6 ^b,c,d^	27.0 ± 6.1 ^a,b^
PLA/MLO-1W-2.5TVS	60.8 ± 0.1 ^b^	106.8 ± 1.2 ^a,b^	170.1 ± 0.3 ^a,b^	14.1 ± 0.4 ^c,d,g^	314.7 ± 4.2 ^b,c^	359.3 ± 1.2 ^c,e,f,g^	3.3 ± 1.2 ^b,c,d^	27.6 ± 6.5 ^a,b,d^
PLA/MLO-5W	60.1 ± 0.3 ^b^	104.1 ± 0.5 ^a,b^	169.4 ± 1.1 ^a,b^	20.2 ± 0.5 ^b,e^	305.7 ± 3.5 ^b,c,d^	361.0 ± 1.0 ^b,c,e,f^	4.1 ± 0.8 ^c,d,e^	29.8 ± 3.5 ^a,b,c,d^
PLA/MLO-5W-1TVS	59.5 ± 0.9 ^b^	105.4 ± 0.2 ^a,b^	168.9 ± 1.2 ^a,b^	16.4 ± 1.7 ^a,g^	316.3 ± 0.6 ^c^	364.3 ± 1.5 ^d^	2.9 ± 0.1 ^b,c^	29.2 ± 1.0 ^a,b,c,d^
PLA/MLO-5W-2.5TVS	59.5 ± 0.3 ^b^	105.9 ± 0.4 ^a,b^	168.5 ± 0.3 ^b^	16.5 ± 0.4 ^a,g^	300.0 ± 3.0 ^d,e^	359.0 ± 1.0 ^a,e,f,g^	4.6 ± 1.4 ^c,d,e^	32.2 ± 2.3 ^b,c,d^
PLA/MLO-10W	60.4 ± 0.8 ^b^	104.1 ± 1.3 ^a,b^	169.8 ± 1.4 ^a,b^	21.4 ± 2.4^b,e^	297.3 ± 4.0 ^d,e^	358.3 ± 0.6 ^a,f,g^	5.3 ± 0.7 ^d,e^	37.8 ± 1.1 ^c,d^
PLA/MLO-10W-1TVS	60.8 ± 0.5 ^b^	107.4 ± 1.7 ^b^	170.4 ± 1.1 ^a,b^	15.8 ± 2.0 ^a,c^	296.7 ± 7.6 ^d,e^	357.3 ± 0.6 ^a,g^	4.6 ± 1.5 ^c,d,e^	36.9 ± 1.6 ^d^
PLA/MLO-10W-2.5TVS	60.4 ± 0.1 ^b^	107.1 ± 2.5 ^b^	169.6 ± 0.9 ^a,b^	15.3 ± 0.7 ^c,d,g^	293.0 ± 5.2 ^e^	360.0 ± 1.0 ^a,b,c^	5.8 ± 0.8 ^e^	36.0 ± 1.0 ^b,c,d^

^a–d^ Different letters within the same property show statistically significant differences between formulations (*p* < 0.05).

**Table 3 polymers-12-02523-t003:** Wettability and CIELab color properties.

	Wettability	Color			
Formulation	WCA (°)	*L*	a*	b*	∆E
PLA	77.2 ± 1.4 ^a^	37.55 ± 0.81 ^a^	−0.19 ± 0.04 ^a^	2.54 ± 0.08 ^a^	-
PLA/MLO	67.7 ± 1.2 ^b^	43.58 ± 0.71 ^b,e^	−1.91 ± 0.10 ^a^	2.38 ± 0.18 ^a^	-
PLA/MLO-1W	83.7 ± 2.1 ^c,f^	47.23 ± 0.34 ^c^	0.48 ± 0.97 ^b^	18.76 ± 0.64 ^b^	17.0 ± 0.7 ^c^
PLA/MLO-1W-1TVS	80.9 ± 1.3 ^d^	51.16 ± 0.46 ^d^	−1.41 ± 0.83 ^a^	15.16 ± 0.17 ^c^	14.9 ± 0.4 ^d^
PLA/MLO-1W-2.5TVS	83.8 ± 1.6 ^c,f^	47.18 ± 1.34 ^c,e^	−0.41 ± 0.85 ^a^	14.19 ± 0.55 ^c^	12.5 ± 0.9 ^e^
PLA/MLO-5W	83.8 ± 1.5 ^c,f^	44.05 ± 0.55 ^b,e^	5.35 ± 1.14 ^c^	22.11 ± 0.43 ^d^	21.1 ± 0.1 ^f,g^
PLA/MLO-5W-1TVS	83.9 ± 2.1 ^c^	44.01 ± 1.54 ^b,e^	5.66 ± 1.09 ^c,e^	21.59 ± 0.99 ^d^	20.7 ± 1.0 ^f^
PLA/MLO-5W-2.5TVS	81.6 ± 0.7 ^d^	46.54 ± 1.01 ^c,e^	4.60 ± 1.09 ^c^	21.49 ± 0.54 ^d^	20.4 ± 0.7 ^f^
PLA/MLO-10W	85.2 ± 0.7 ^e^	37.73 ± 1.78 ^a^	8.60 ± 1.02 ^d,e^	19.46 ± 1.42 ^b^	21. ± 0.8 ^f,g^
PLA/MLO-10W-1TVS	81.4 ± 1.7 ^d^	45.04 ± 0.98 ^e^	7.86 ± 1.46 ^d,e^	22.82 ± 1.32 ^d,e^	22.7 ± 1.7 ^g,h^
PLA/MLO-10W-2.5TVS	83.3 ± 2.1 ^f^	45.24 ± 0.35 ^b,c,e^	7.51 ± 1.07 ^e^	23.81 ± 0.41 ^e^	23.5 ± 0.1 ^h^

^a–h^ correspond to Figure 11.

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
