# Peer review of "Silane-Functionalized Sheep Wool Fibers from Dairy Industry Waste for the Development of Plasticized PLA Composites with Maleinized Linseed Oil for Injection-Molded Parts"

_polymers, 2020, doi:10.3390/polym12112523_

Round 1

Reviewer 1 Report

The authors report poly(lactic acid)/maleinized linseed oil composites reinforced with sheep wool fibers. Furthermore, wool fibers were firstly functionalized with 1 and 2.5 phr of tris(2-methoxyethoxy)(vinyl) (TVS) silane coupling agent. This work is based on the article Effect of Different Compatibilizers on Injection-Molded Green Fiber-Reinforced Polymers Based on Poly(lactic acid)-Maleinized Linseed Oil System and Sheep Wool.

According to the authors, this new work aims to improve the compatibilization of the fiber with the composite. However, they only propose a concentration of silane. The authors would have studied different silane concentrations and their effect on mechanical and thermal properties, based on experiment design. Typically, when there is excellent compatibility, some property increases or is improved. Is the silane concentration of 2.5 phr the best?

Another recommendation is to determine the amount of functionalized silane on the fibers by TGA in the oxidative atmosphere, see reference 28, 42.

Molecular vibration at 763 cm-1, what does it correspond to this peak? Could you determine the degree of functionalization with this peak?

Figure 5a needs to be modified; the legend is missing on the Y-axis.

Is there a chemical reaction between the functionalized fiber with silane and maleinized linseed oil during the extrusion process?

Concerning the introduction, can this be a little shorter?

Author Response

Reviewer 1

The authors report poly(lactic acid)/maleinized linseed oil composites reinforced with sheep wool fibers. Furthermore, wool fibers were firstly functionalized with 1 and 2.5 phr of tris(2-methoxyethoxy)(vinyl) (TVS) silane coupling agent. This work is based on the article Effect of Different Compatibilizers on Injection-Molded Green Fiber-Reinforced Polymers Based on Poly(lactic acid)-Maleinized Linseed Oil System and Sheep Wool.

We thank Reviewer 1 for his/her valuable comments as well as for considering our manuscript suitable for its publication in Polymers.

According to the authors, this new work aims to improve the compatibilization of the fiber with the composite. However, they only propose a concentration of silane. The authors would have studied different silane concentrations and their effect on mechanical and thermal properties, based on experiment design. Typically, when there is excellent compatibility, some property increases or is improved. Is the silane concentration of 2.5 phr the best?

In fact, we studied two different concentrations 1 and 2.5 phr and their effect on mechanical and thermal properties, based on experiment design. In the current version of the manuscript we have analyzed the thermal stability (by isothermal TGA at the processing temperature of the biocomposites) as well as the crystalline structure (XRD) of the untreated fibers as well as the silane treated fibers. From the overall results, the silane concentration of 2.5 phr is considered the best one. In fact, the surface silane treated wool fibers at the highest concentration of 2.5 phr of TVS, at loading amounts of 1 phr and 5 phr into PLA/MLO matrix (PLA/MLO-1W-2.5TVS and PLA/MLO-5W-2.5TVS), lead to the composites with the highest Young's modulus and elongation at break, show enough thermal stability for the processing conditions as well as for the intended use, and showed improved surface water resistant.

Another recommendation is to determine the amount of functionalized silane on the fibers by TGA in the oxidative atmosphere, see reference 28, 42.

Thank you for this suggestion. In the current version of the manuscript, the TGA measurements were performed under isothermal conditions at the processing temperature of the composites under air conditions (see Figure 4). Although, the TGA test could not lead to determine such low amount of functionalized silane, TGA isothermal analysis was able to showed the improved thermal stability of fibers.

Molecular vibration at 763 cm-1, what does it correspond to this peak? Could you determine the degree of functionalization with this peak?

The peak at 763 cm-1 corresponds to Si-C bond showing presence of silane. As its shift into lower wavelength values indicates coupling effect. However, the intensity of the mentioned peak couldn`t be used to determine degree of functionalization. Nevertheless, in the current version of the manuscript we have extended the discussion of FTIR results and we believe that it is more clear as well as it has been substantially improved.

Figure 5a needs to be modified; the legend is missing on the Y-axis.

Thank you for the suggestion. Figure 5 has been corrected in the current version of the manuscript.

Is there a chemical reaction between the functionalized fiber with silane and maleinized linseed oil during the extrusion process?

No chemical reaction is expected during extrusion process. Although double bond (C=C) is present after silane functionalization of sheep wool, there is no intended free radical in MLO that allows to make the reaction during the extrusion process.

Concerning the introduction, can this be a little shorter?

The introduction has been reordered and shortened.

We believe that now the manuscript have been improved and we wish that it is now suitable for its publication.

Reviewer 2 Report

It's a good work to describe new nanofibers in PLA-based materials, in which the methods were detailed and specific. Although the thinking and methods were normal as other works showing, it could tell some new information about how this new composites displayed structures and properties.

I would recommend that with TEM/XRD results this work would be more appropriate for publishing,

Author Response

Reviewer 2

It's a good work to describe new nanofibers in PLA-based materials, in which the methods were detailed and specific. Although the thinking and methods were normal as other works showing, it could tell some new information about how this new composites displayed structures and properties.

We thank Reviewer 2 for his/her valuable comments as well as for considering our manuscript suitable for its publication in Polymers.

I would recommend that with TEM/XRD results this work would be more appropriate for publishing,

Thank you for this recommendation. The TEM analysis was not performed since we have observed the microstructure of sheep wool fibers by SEM where is possible to oberve the fiber cuticle cells adhered in unfunctionalized fibers, while in functionalized sheep wool fibers the detachment of cuticle cells increased.

The XRD of  sheep wool fibers (Figure 4-b) as well as of composites (Figure 8) were added in the current version of the manuscript. Correspondingly, the discussion of these results have been added.

We believe that now the manuscript have been improved and we wish that it is now suitable for its publication.

Round 2

Reviewer 1 Report

The authors attended the comments made; however, the manuscript still presents errors. For example:

Please change the word wavelength by wavenumber from the next paragraph:

The absorption band at 763 cm-1 corresponds to the –Si–C– symmetric stretching bond [52], and as it was shifted to lower wavelength values (760 cm-1) in the silane treated fiber with 2.5 phr of TVS it may indicates coupling effect.

Please highlight or point the peaks to 1023 and 840 cm-1 in Figure 3 for the TVS sample. The quality of the figure is not good.

Please correct references 5, 25, 27, 53 are incomplete.

Author Response

The authors attended the comments made; however, the manuscript still presents errors. For example:

Thank you for your suggestions.

Please change the word wavelength by wavenumber from the next paragraph:

The absorption band at 763 cm-1 corresponds to the –Si–C– symmetric stretching bond [52], and as it was shifted to lower wavelength values (760 cm-1) in the silane treated fiber with 2.5 phr of TVS it may indicates coupling effect.

The word wavelength have been changed by  wavenumber.

Please highlight or point the peaks to 1023 and 840 cm-1 in Figure 3 for the TVS sample. The quality of the figure is not good.

The peaks have been highlighted in Figure 3. The quality of the figure have been improved and it was provided separately to the editorial office to be further included with high quality in the final version of the manuscript.

Please correct references 5, 25, 27, 53 are incomplete.

The references have been completed.